# Mutations in SARS-CoV-2 are on the increase against the acquired immunity

**Tomokazu Konishi** *

Graduate School of Bioresource Sciences, Akita Prefectural University, Akita, Japan

* konishi@akita-pu.ac.jp

## Abstract

Monovalent vaccines using mRNA or adenoviruses have provided substantial protection against the COVID-19 pandemic in many countries. However, viral mutations have hampered the efficacy of this approach. The Omicron variant, which appeared in Dec 2021, has caused a pandemic that has exerted pressure on the healthcare system worldwide. The COVID-19 vaccines are not very effective against this variant, resulting in an increased rate of infection and mortality. Owing to the rapidly increasing number of patients, few countries, such as Australia, New Zealand, and Taiwan, which aimed at zero-COVID cases, have discontinued their attempts to contain the spread of infection by imposing strict lockdowns, for example. Therefore, the administration of booster vaccinations has been initiated; however, there are concerns about their effectiveness, sustainability, and possible dangers. There is also the question of how a variant with such isolated mutations originated and whether this is likely to continue in the future. Here, we compare the mutations in the Omicron variant with others by direct PCA to consider questions pertaining to their evolution and characterisation. The Omicron variant, like the other variants, has mutated in humans. The accumulated mutations overwhelmed the acquired immunity and caused a pandemic. Similar mutations are likely to occur in the future. Additionally, the variants infecting animals were investigated; they rapidly mutated in animals and varied from the human strains. These animal-adapted strains are probably not highly infectious or pathogenic to humans. Hence, the possibility of using these strains as vaccines will be discussed.

**Data Availability Statement:** https://doi.org/10.6084/m9.figshare.19029653.v1

**Funding:** The authors received no specific funding for this work.

**Competing interests:** The authors have declared that no competing interests exist.

## Introduction

The COVID-19 pandemic has continued despite the efforts of many countries to contain it [1]. It is thought to have started in Wuhan, China. Then, by April 2020, it had spread to Europe and North America. During this progression, it rapidly mutated to form four major sub-groups, three of which are still prevalent today [2]. COVID-19 is also known to have spread among several susceptible animal species; a problem that, as in the case of humans, continues to manifest itself [3–6]. Subsequently, increased surveillance at national borders has slowed the spread of the disease across national lines. Further, more potent variants have emerged in each country independently [7]. The most infectious sub-types have spread across borders and have

been designated as variants of concern (VOC) or variants of interest (VOI) by the WHO [8]. For easy interpretation, these variants are represented with Greek letters beginning from Alpha, and this has been followed throughout the manuscript.

Countries have taken measures to prevent the spread of the disease by surveying and isolating the infected people. Vaccines have been rapidly developed; particularly monovalent vaccines. This has been made possible through the use of new technologies, such as RNA vaccines, that have become popular globally. These have proven very effective, and have led to a significant reduction in the number of people infected for a time; even in countries where detection and isolation did not work well [9–21].

However, as the virus continues to mutate, the effectiveness of the vaccines continues to wane, and the Delta variant is emblematic in this situation. More recently, infection by the Omicron sub-type has escalated [22–25]. Even in Australia, where vaccination rates are high and effective control measures are in place, the Omicron variant has caused many cases of infection [26]. This is probably because the virus mutated much faster than the vaccine could update; the virus is likely to be infectious even after the third dose of the vaccine [27]. In Japan, for example, vaccination history is unrelated to the number of new positive cases [11]. Additionally, vaccines do not appear to be effective in preventing long-COVID, i.e., an illness with very long-lasting symptoms [12–14]. Booster doses of vaccines, however, are thought to be effective in preventing severe disease [15]. Furthermore, we are warned in advance that mRNA vaccines have side effects such as fever; actually, this is a frequent adverse reaction. However, more and more serious adverse reactions are being reported. These include the incidence of shingles [16–20]. Guillain-Barré and fulminant hepatitis, which are often reported with other vaccinations, may also occur with this vaccine [21,28].

Here, the Omicron variant's genetic sequence was characterised by using direct principal component analysis (PCA) [29] and discussed with the mechanism informing its virulent manifestation. This is an objective method to evaluate the characteristics of a sample based on sequence differences; with each PCA axis presenting differences in the nucleotide sequences at specific positions. In this analysis, several axes are used to determine factors, such as the origin of each variant, how it has changed, and its basic characteristics. This data will be compared with that obtained from variants infecting animals to discuss the possibility of developing a vaccine using a weakened variant of the virus.

## Materials & methods

The FASTA package of all the nucleotide sequences was downloaded from GISAID [30] on 27 December 2021. However, the set did not include samples from African countries other than South Africa. Thus, to increase the number of African samples, those with complete sequences from 1 July 2021 to 15 January 2022 were also downloaded. Only the complete sequences that contained less than 1,000 N were selected. The sequences were aligned using the DECIPHER [31]. Subsequently, they were converted to a Boolean vector and subjected to PCA [29]. Sample and sequence PCs were scaled based on the length of the sequence and the number of samples, respectively [32].

The PCA axis shows differences in a specific set of bases. The axis is determined using our designated search dataset. Therefore, depending on the set of samples used, the observed differences will vary. Depending on one's aim, there are several viable sets of axes available. One is the initial axes on human acclimatization, which was created using data up to April 2020, and spread radially across four groups, and was used to determine variant origin. The other axis was derived using each of the two WHO-VOC and VOI [8], Alpha to Omicron. This is because the data were not balanced since a particular variant caused a significant number of

infections. The bias in the weight of such unbalanced data can skew the results of PCA. In this axis, the most highly mutated Omicron variant formed PC1; this was used to determine the variation in the Omicron variants. The remaining variants were divided in PC2. In addition, to characterise the samples infecting animals, 1500 samples and the two WHO-VOC were used.

All calculations were performed using R [33]. The ID, acknowledgements, list of samples used for the WHO-VOC, PCA axes, and scaled PCs of samples and bases can be downloaded from Figshare [34]. The newest version of the R code is publicly available at GitHub [35].

## Results

First, the effectiveness of the vaccine was checked by using the data on the number of people infected in Japan. Although the fifth pandemic appeared to be over after the first and second round of vaccinations (green), the sixth wave occurred, leading to the most extensive pandemic (blue in Fig 1A). A certain percentage of the infected individuals required hospitalization, as observed in waves 1–5 (Fig 1B), but the number of people with severe symptoms was lower than expected (Fig 1C). There was a large flicker in the relationship between the number of positives and the number of deaths, perhaps because it reflects the lethality of the various variants. Although the number of deaths in the sixth wave was lower than expected (Fig 1D), this decrease was not as much as the number of people with severe symptoms (Fig 1C).

Next, the transition of the variants was explored by using PCA. When PCA axes were evaluated from the data up to April 2020 and compared to the recent data on these axes, the variants appeared as several groups on three routes (Fig 2A) [2]. At this stage, the virus's acclimation to humans seems to be complete. The changes made here are of great importance, and this is probably why the current variants retain the same characteristics. In Fig 2A, the axis shows 27,000 random samples of variants registered up to 27 December 2021. In blue is the VOC of the WHO. All the Omicron variants belonged to group 1.

Incidentally, approximately 1500 sequences of the SARS-CoV-2 variants infecting animals have been registered as of 27 December 2021. Notably, they too belong to one of these mentioned groups (Fig 2B). In particular, many of the sub-forms prevalent in minks, deer, dogs, cats, and zoo animals are thought to have been transmitted by humans; specifically, they likely originated from variants where many human cases have been observed.

The currently prevalent variants have many more mutations. Fig 3A reflects the magnitude of mutations in the variants. To equalise the weights, two WHO-VOC each were selected to set the axes. The Omicron variants were observed to be distant from the others. The samples from African countries recorded changes in the variant. The number of reported cases in the upper right corner increased, suggesting that they became more infectious. The upper rightmost variants have a three-amino acid insertion in the spike protein sequence. This is noteworthy because, while many of the newer variants have some deletions, insertions are rare. For example, as deletions in spike, Alpha has 3 amino acids, Beta and Delta have 2 amino acids, and Omicron has 6 amino acid deletions. Also in NSP6, Alpha, Beta, and Omicron have a common cohesive 3 amino acid deletion. In contrast, only some Omicron variants have the three amino acid insertion on the spike. Thus, the variants with the insertion are probably those that have mutated the most among Omicrons. Incidentally, the WHO now classifies Omicrons from BA1 to BA5, but this insert has nothing to do with this classification; part of BA.1 and part of BA.2 each have this insert. It should be noted that the spread of mutations is not the process of change observed on a time-series basis. The first two reported cases in South Africa were already heavily mutated (10/12 and 10/24). The earliest Omicron variant, which was still less mutated, would have been located farther down to the left. It is likely that the disease spread

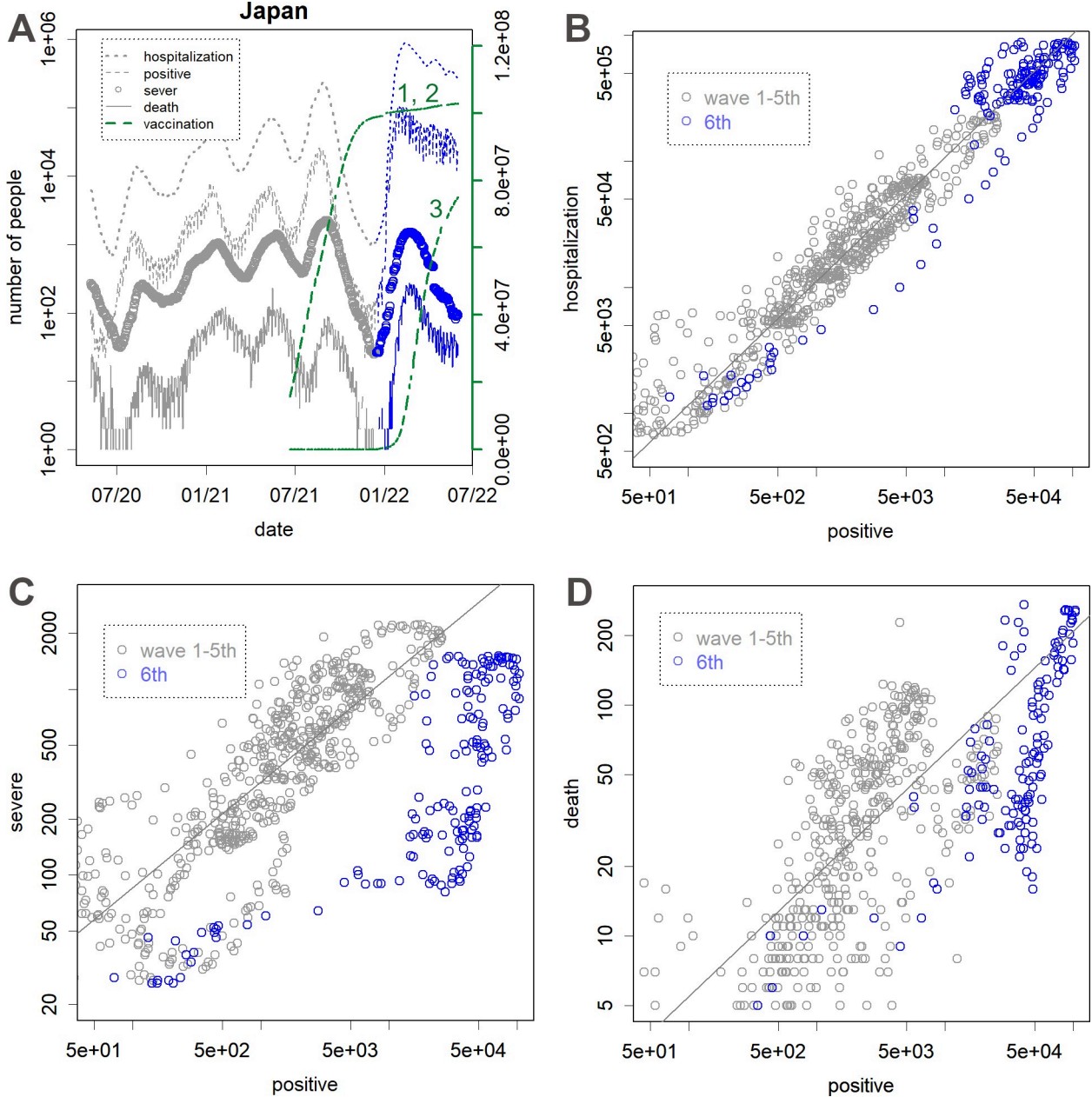

**Fig 1. Changes in the number of patients [36] in Japan. A**. From the top, the number of hospitalized patients, new patients with positive PCR test on that day, severe patients requiring intensive care, and deaths. The blue lines represent the 6th wave. The green lines show the 2nd and 3rd rate of vaccinations (on the right y-axis). **B**. The number of newly positive patients vs. hospitalized patients. **C**. The number of newly positive patients vs. patients with severe symptoms. **D**. The number of newly positive patients vs. deaths (2 weeks later). Fit lines (grey) for 1st–5th waves were obtained by a robust method, i.e., the *line* function of the R.

elsewhere, matured, and then the most prevalent variant moved to the sequencing countries such as South Africa (Figs 3B and S1).

The global data in Fig 3A are shown over time (Fig 3C and 3D). It can be seen that a single epidemic in each region was caused by a single variant, where a change in the variants was discontinuous. The gap to the Omicron variant is emphasized by the absence of sufficient African records. This is distinctly different from the case of H1N1 influenza. If the mutations were to

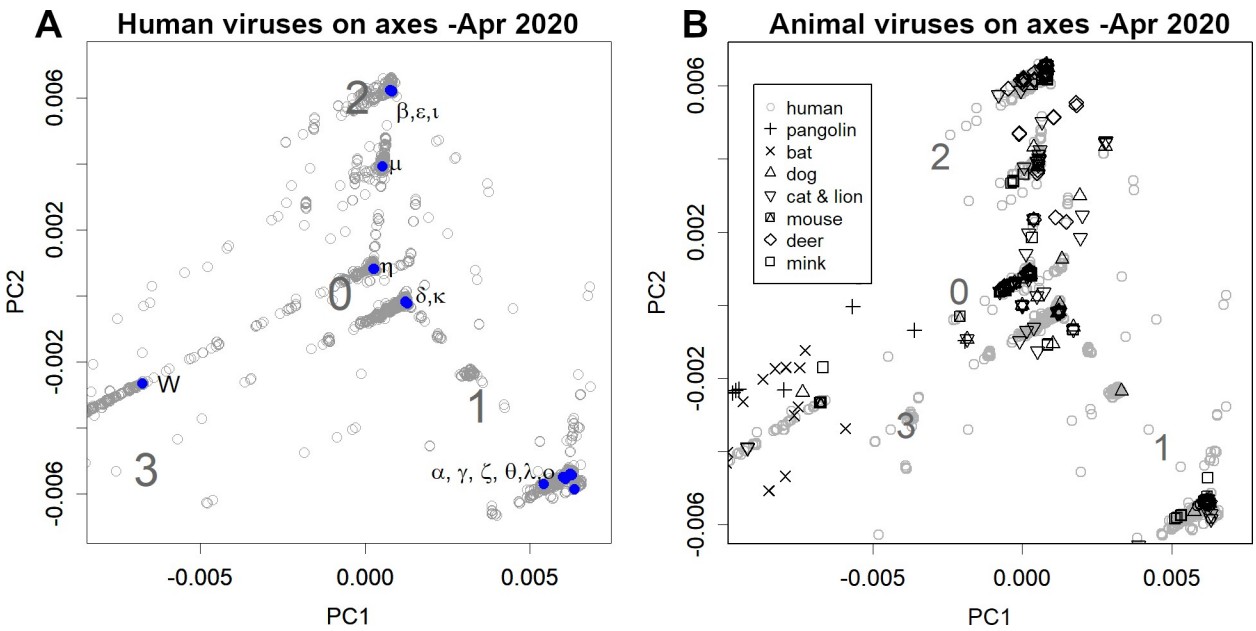

**Fig 2. Principal component analysis (PCA) with axes found in the data up to April 2020.** The axes reflect the differences in the data at this point. **A.** These data, in addition to 27 000 randomly selected human samples from the data up to December 2021 have been shown. Blue is WHO-variant of concern (VOC); W is the first variant. **B.** Animal samples are shown on the same axis as in A.

accumulate sequentially in one variant, PCs would show sine curves, as seen in H1N1 mutations (S2 Fig, the variant repeatedly caused a pandemic from the 1970s to 2009. The same phenomenon was also found in pdm09 [37]). There was one variant of H1N1 per year somewhere in the world, which moved annually while changing itself. After a few years, the variant would change by approximately 15–30/1000 bases and then return to the same location to cause another epidemic. The reason the same variant did not cause an epidemic the following year was because those that were infected by the variant in the population gained an acquired immunity against it without developing the disease. If the variant does not mutate sufficiently to overwhelm the immunity, it cannot cause an epidemic again [37].

Omicron was first reported in South Africa; however, Group 1 variants to which Omicron belongs were not prevalent in this country after August 2020 (Fig 3B). The only Group 1 variant that appeared briefly in July 2021 was C.1.2, which is also quite far from the Omicron variant (Fig 3B). A closer group 1 variant is B.1.1.519, which was reported by Botswana and Morocco. The relationship between this variant and Omicron and its origin remains unknown because of lack of records, as will be discussed later.

Omicron is a mutated human variant of COVID-19. However, this variant's mutations did not resemble any of the existing coronaviruses (Fig 3B and 4A) [38], nor did it have anything in common with SARS-CoV-2 that had infected animals (Fig 4C). Thus, this eliminated the possibility that it was transmitted from animals [39]. In particular, the rodent data were completely unrelated to Omicron's mutations (S4 Fig). This animal vector hypothesis originally arose as a result of processing the phylogenetic results with PCA. However, phylogenetic trees are a form of one-dimensional data created based on the distances between sequences. Therefore, these sequences are not comparable. Further, given that PCA is a method for observing multidimensional data, processing one-dimensional results is not its original purpose. Artefacts caused by inappropriate data processing were apparently the source of this concern. As seen in Figs 2A and 3A, this variant gradually mutated within group 1.

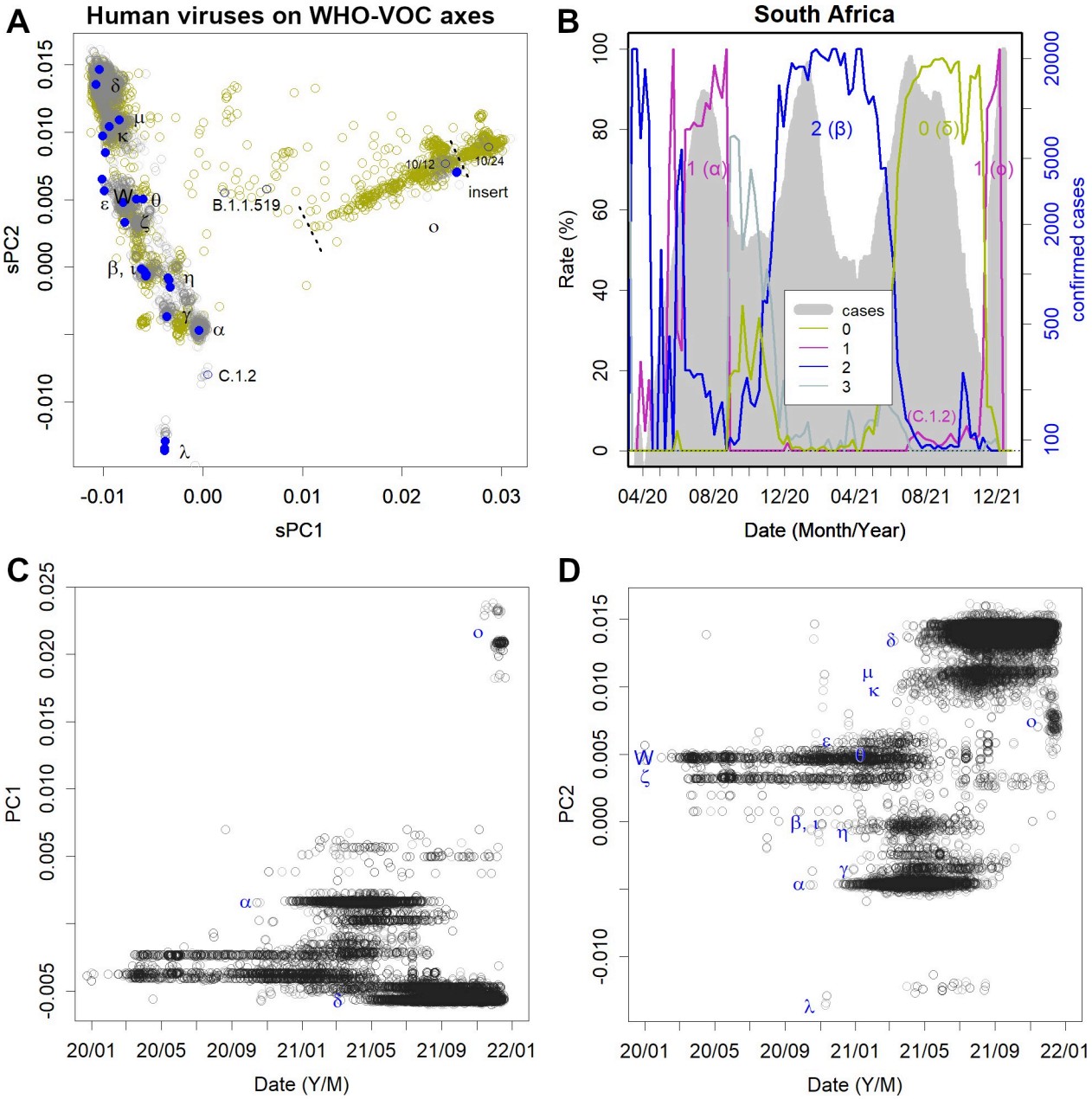

**Fig 3. Spread of Omicron variants. A**. Human data is presented the axes determined by WHO- variant of concern (VOC). Each axis reflects the differences among these variants. Grey reflects 27 000 randomly selected data from the global data. Khaki reflects African data of 17 000 samples from July 2021 to January 2022. Omicron is depicted on the right-hand side of the figure; 10/12 and 10/24 were the first reported cases in South Africa (blue). **B**. Pandemics in South Africa (grey) and groups accounting for the percentage of cases at each time point (coloured lines) are specified. **C, D**. The time course of the global data is presented in A. Each variant has a range of mutations depending on the number of patients but does not change continuously. Rather, another, more potent variant creates the next pandemic.

When SARS-CoV-2-infected animals, such as mink, deer, dogs, and cats, a ping-pong effect occurred, thereby increasing the number of infected animals. In these animals, acclimatisation occurred quickly. This is similar to the situation in which the initial SARS-CoV-2 variants were acclimatised to humans by April 2020. For example, mutations in PC21 and PC25 (Fig 4D) on the animal sample axis suggest acclimation to minks and deer in some countries.

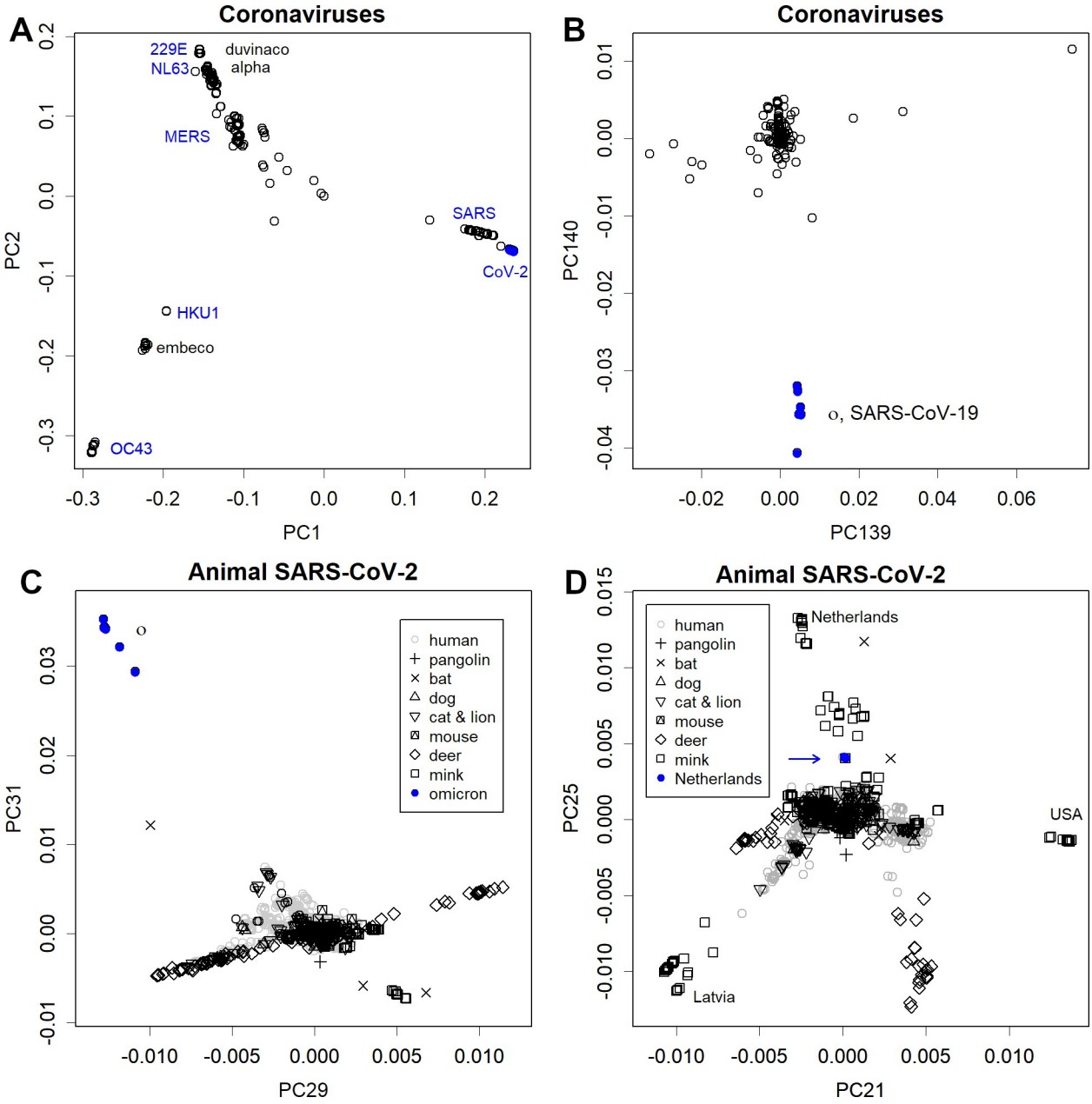

**Fig 4. Results for the axes found in the various samples. A**. Axes found in coronaviruses. All SARS-CoV-2 are on the far right. **B**. The same series of axes of principal component (PC) 139 and 140, showing the characteristics of Omicron variant. The fact that all coronaviruses do not appear in this neighbourhood indicates that Omicron has features completely independent of other variants. **C**. Axes found in animal SARS-CoV-2 and WHO-variant of concern (VOC). There is no data in the vicinity of the omicron, indicating that the causal mutations were unique. **D**. PC 21 and PC 25 of the same set of axes as C, showing the characteristics of mink and deer variants in several countries. Blue arrow shows the mink variant prevalent in humans.

The concern about re-infection from these animals to humans is natural. However, variants that are sufficiently far from the human variants, as shown in Fig 4D, did not re-infect humans. This is why 27,000 human samples are clustered in the centre. If massive re-emergence should occur in the future, it would be easily confirmed by sequencing. In fact, the only variant that has ever been prevalent in humans is the one in the Netherlands, indicated by the blue arrow.

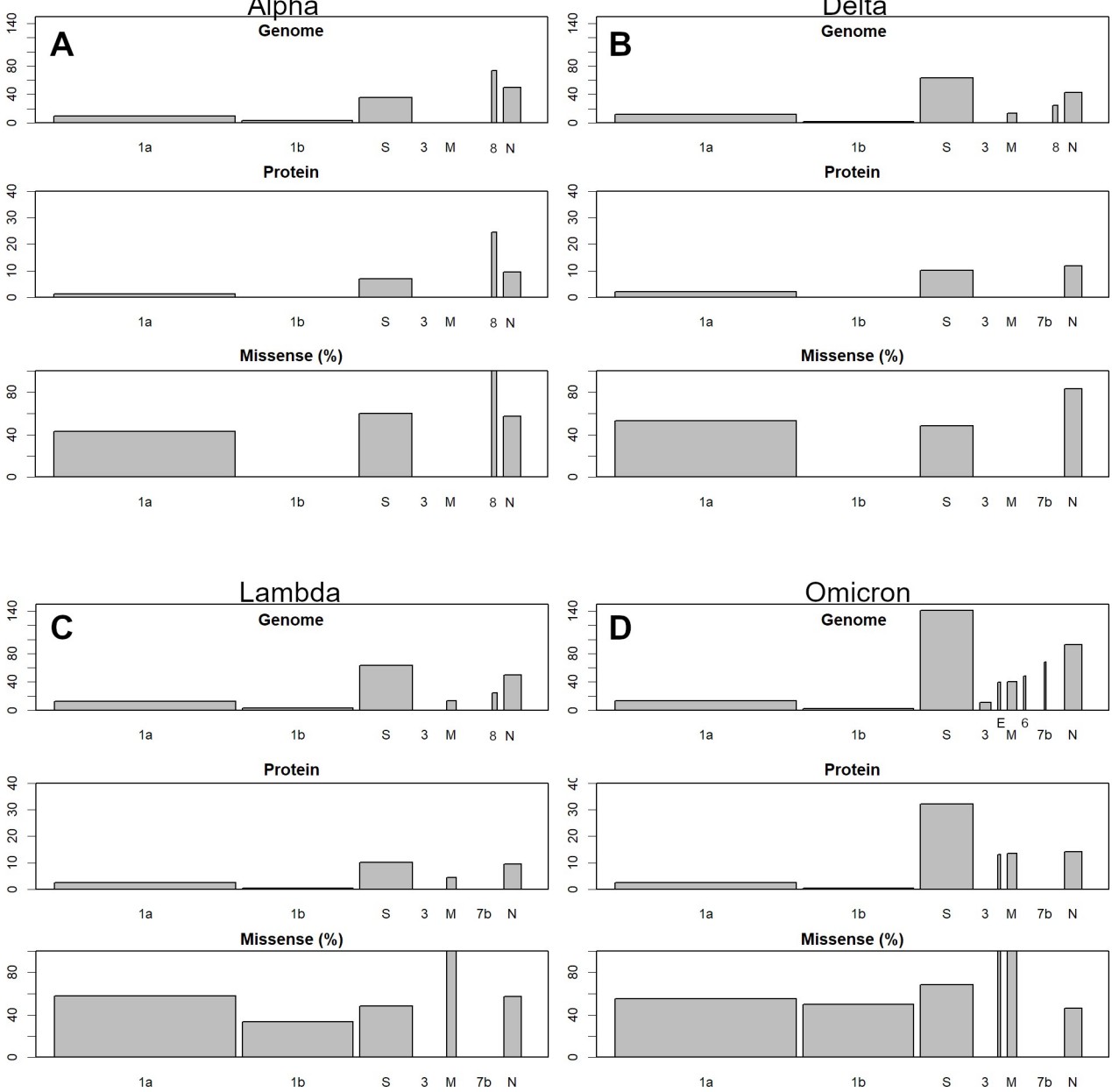

**Fig 5. Changes in each variant compared to average data as of April 2020.** Each panel consists of three sub-panels; from top to bottom: Genomic variation, protein variation, and the proportion of missense mutations. The top two panels show the number of mutations per 1000 bases or residues, with S and N being the most prominent: **A**, Alpha; **B**, Delta; **C**, Lambda; and **D**, Omicron. Omicron variant has a particularly high number of S protein mutations.

This variant is far from human viruses, but it is even farther from the mink viruses. Thus, it is probably the process of acclimation to the mink. During the pandemic phase dominated by this variant, the mortality rate in the Netherlands declined by approximately the same number of patients [40].

The mutations occurred mainly in spike glycoprotein (S) and nucleocapsid phosphoprotein (N) (Fig 5). This is very different from influenza, in which all ORFs change simultaneously at the same rate [37]. The mutations in Delta variant are larger than those of Alpha. Further,

since these are opposite mutations across the initial variant (Fig 3A), Delta would have been spared much of the immunity gained by Alpha. For example,

```
            ***.*******
Alpha     CTCATCGGCGG
Earliest  CTCCTCGGCGG
Delta     CTCGTCGGCGG
```

Here, part of the sequences from 23610 to 23620 is shown with the earliest variant. For 23613, base A has a negative sPC_base value. As the Alpha variant has more such negative mutations, it is negative in the sPC_sample (Fig 3A), and vice versa. Lambda has more mutations than these, with Omicron having even more of them. The mutations mainly occur in S and N, which are the surface and nucleocapsid proteins of the virus, respectively; therefore, there must be a strong selection pressure to overcome immunity [41]. In Omicron, there was a high density of S mutations suggesting that there was selection pressure to avoid the acquired immunity imparted by monovalent vaccines. In Omicron, the mutations are also in the smaller ORFs, which are relatively well preserved. The mutation in the envelope (E) is only one amino acid, but it is very rare. In addition, there are three amino acid mutations in M (Fig 5D). These mutations appear particularly at PC_base. Since an Omicron sample has a high positive sPC1 value (Fig 3A), many, if not all, bases in PC_base1 with a strong positive value would be found in Omicron. The sequence of each sample should be checked individually, since sPC_sample depicts the overall scenario.

The animal viruses did not show the same concentration of S and N mutations as human viruses, for example, Alpha. Fig 6 shows the number of mutations of the variants that vary from that of the virus infecting the human population (Fig 4D). There were more missense mutations; therefore, some amino acid mutations may have been desirable for each host's specificity. However, many small ORFs were retained, and none of them caused major mutations, such as seen in Fig 5. This does not necessarily mean that variants that are more acclimated to humans are less likely to infect animals, but the examples shown here were relatively early in the process of infecting animals incidentally (so they would have had more time to get away from humans). Other variants, for example, delta and lambda, can also infect animals (S5 Fig).

## Discussion

Omicron did not arise in South Africa. Specifically, the parent of this variant was not prevalent in South Africa. Rather, it probably originated in areas without sequence testing, matured sufficiently to overcome the vaccine-acquired immunity and then entered the sequencing countries. By the time the danger was recognised in the South African survey, the variant had probably already spread to other parts of the world. The current global pandemic may be the result of this delay.

The Omicron variant has affected many individuals in most countries, including Japan, due to its high infectivity (Fig 1A). Unfortunately, the size of the infected population is large because the vaccine has little effect on the infection (Fig 1A) [11] and on reducing hospitalizations (Fig 1B). Although the number of severe cases reduced (Fig 1C) [15], it is unclear whether the reduction was due to the vaccination, as this was not observed with earlier variants. Rather, it is possible that the Omicron variant has the tendency to invade organs other than the lungs, which appeared to be the primary target and therefore were the first to be

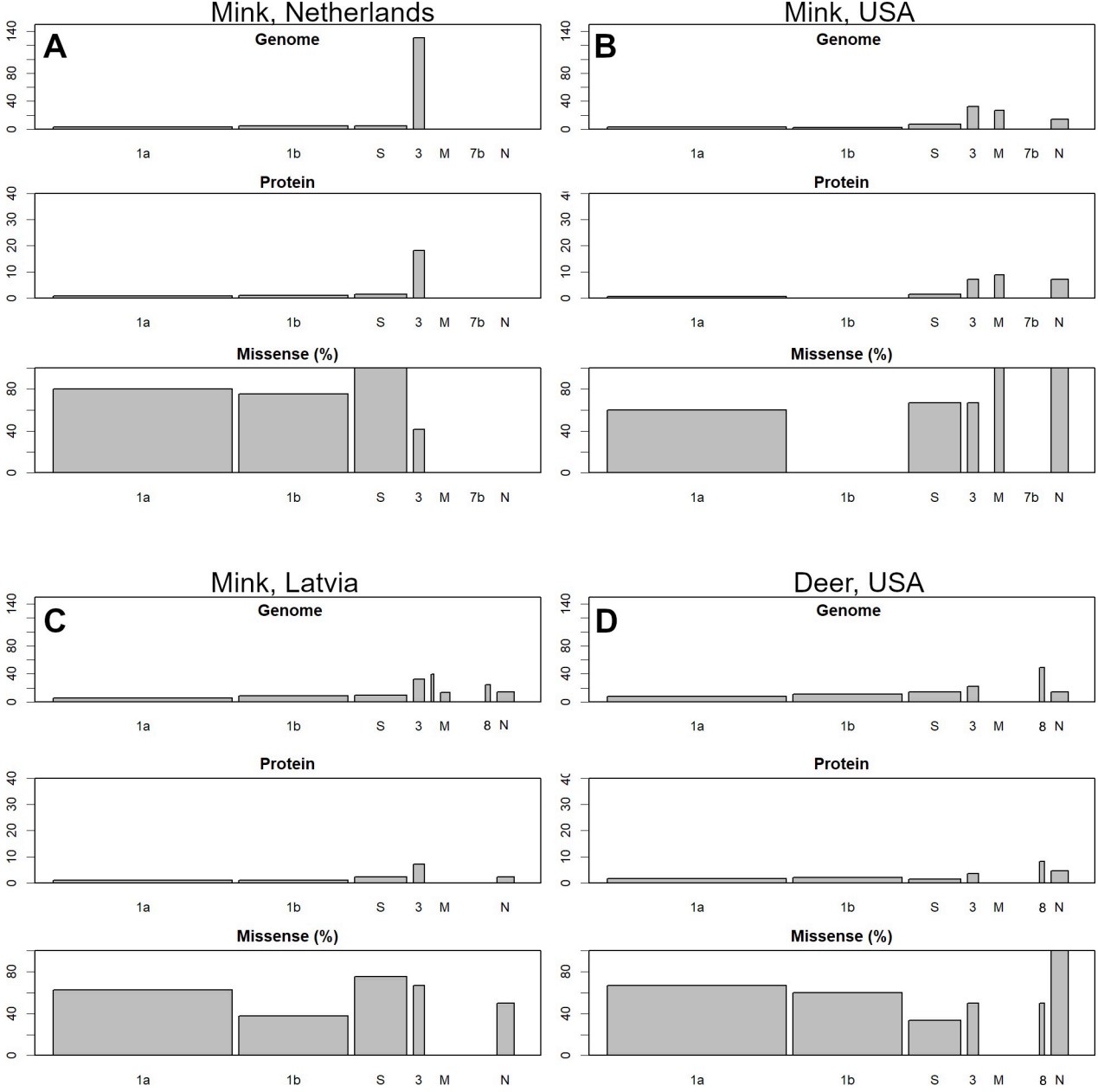

**Fig 6. Animal variants compared to the average human data up to April 2020.** Those are the most distant variants found in Fig 4D. The mutations are smaller than those of WHO-variant of concern (VOC) in Fig 5 and are not concentrated in the S or N.

examined while detecting severe cases. The number of deaths also reduced; however, the reduction was not as much as that of the severe cases (Fig 1D), mainly because of the loss of people who were not identified with severe symptoms [42]. These findings suggest that the early mRNA vaccine has lost its effectiveness. Accordingly, the sixth peak in Japan is becoming extremely high without subsiding, which can be due to dependency of the government only on the vaccines [43].

In contrast to the influenza H1N1 virus, in which a single variant changed gradually [37], SARS-CoV-2 has many variations that continue to change in different ways (Figs 1, 2 and S1). This is because there were three groups in the early stages that evolved independently, in

different regions, and after the borders were closed, and the evolved stronger infectious variants were successively released on a global scale [2]. Even a variant as infectious as Delta, for example, does not infect everyone; this is because people are consciously protecting themselves. However, if a new, more infectious variant arises, it can break through these artificial defenses. It is also possible for a very different variant to overcome acquired immunity. With the widespread use of monovalent vaccines, many people are now immune to certain variants. Omicron has been able to evade this immunity and has spread the disease due to high variations from previous variants.

Africa is home to 1.2 billion people, but there are few areas where sequencing is routinely performed. The number of sequences per population in Africa was only 1/150 of that in Europe (S3 Fig), and 40% were from South Africa. Hence, there is a relative dearth of records compared to other regions. This is also the case in many Asian and Latin American regions. Similar gaps in the records can be seen in the H1N1 influenza viral mutations [37]; which mutate continuously every year, but still sometimes reveal large gaps. For example, the variant that was prevalent from the 1970s to 2009 had three gaps before and after 1985, 1996, and 2002 [37]. Thus, the gaps in Lambda and Omicron (Fig 3A) are likely due to this lack of records.

The number of variant sequences from the USA and UK was exceptionally large. However, considering the COVID-19 situation in these countries, it seems that their huge amount of sequencing is not doing much to prevent the spread of infection. If these countries had been more generous lending some of their capacity for sample sequencing to developing countries, they would have been able to detect the new variants more quickly. If detection had occurred at an earlier stage, quarantine could have stopped the spread. Thus, there is a need for international cooperation to conduct such surveys.

Apparently, the PANGO nomenclature system is still analysing the data by using a phylogenetic tree. This has led to rare genetic events, such as insertions, into the Omicron variant being ignored. Although a phylogenetic tree has been used for a long time, it is not possible to show the relationships between sequences and clusters as in the PCA method. Moreover, it is difficult to ensure objectivity [29]. In this situation, however, it is challenging to follow the transition of the variants.

Monovalent vaccines have been used to combat the COVID-19 pandemic. These targeted the S-protein and worked well, but the Omicron variants were more capable of evading this immunity. For this reason, many countries and regions are rushing to grant booster vaccinations. However, repeated vaccinations may not be sustainable [44–46]. In fact, in many areas, even the first round of vaccination has not been completed [47]. With regard to Israel, the effectiveness of boosters is said to be questionable [48]. In fact, there is a report that repeated boosters do not work [27]. There have also been concerns about the dangers of repeated booster doses that could eventually weaken the immune response and exhaust people [49]. The likelihood of adverse reactions, such as the incidence of shingles [16–20] or Guillain-Barré [21,28], might increase with repeated booster doses. Therefore, quarantine based on availability of monovalent vaccines must be revised.

I wish to point out the possibility of using animal-adapted variants to develop a multivalent SARS-CoV-2 vaccine, such as that for the vaccinia virus for smallpox. Live vaccines show a significantly higher immunogenicity than inactivated vaccines because natural infection is better imitated and a wide range of immunologic responses are elicited [50]. However, the main disadvantages of live vaccines are safety concerns. In fact, a half-adapted mink variant was barely able to spread among humans. It probably had low virulence and was quickly replaced by a more infectious variant. A more adapted variant would probably not be able to spread from humans to humans. Once a weakly toxic variant is selected, it can be maintained and propagated in its host and cultured cells. The efficacy can be expected from the fact that SARS-CoV-

2 does not mutate, particularly small ORFs. Perhaps the virus does not have sufficient flexibility. However, in the body, all proteins could be presented as antigens. This is why all the ORFs were altered in the influenza virus and this virus has been prevalent for decades [37]. Such viruses may be less effective in preventing infection than RNA vaccines targeting the S protein. However, they are more resistant to S protein mutations and may hold the potential for preventing severe symptoms.

The H1N1 influenza haemagglutinin mutated and replaced most of the protein's surface between the 1970s and 2009 [37]. If similar degrees of freedom exist in the S and N of SARS-CoV-2, then these should still have a high mutation potential. Omicron did not simply have many variations. For example, the spike gene in EPI_ISL_8038656, BA.1, differs from the reference WIV04 sequence, the earliest variant, by 23 amino acids. On the other hand, EPI_ISL_6795842 (BA.1) contains all of these, but with 35 more substitutions and an insert of 3 more amino acids. This would show that there is more room for mutations and they continued to mutate just like the other variants. Of course, the more people infected the more variations would occur. It is very important to stop the pandemic in each country so that we do not have another VOC. Hence, this effort must be coordinated on a global scale. The production and transportation of weaker variants are much lower-tech than RNA vaccines and are probably more sustainable. If a new RNA vaccine becomes available for Omicron, a mutation may occur that overcomes the newly-developed immunity and causes the next pandemic. If this cycle repeats itself, SARS-CoV-2 may continue to change in a discontinuous fashion. This is a calamity that is difficult to control and will take many years to overcome. If, on the contrary, a multivalent vaccine is approved for practical use, the selective pressure would not be concentrated on the spike protein (S), even if SARS-CoV-2 continues to mutate, similar to influenza viruses. In this case, the pandemic will probably be small, making it possible to relax preventive measures.

## Conclusion

SARS-CoV-2 has many variations that continue to change. Omicron is one such variant generated in countries where the sequences were rarely checked. There are marked differences in the number of sequences reported by countries; for example, the number of sequences reported in Africa is very small. The Omicron variant negatively affected many patients, since the mRNA vaccines have limited effects in preventing it. To prevent the continuous increase in the number of new variants, it is important to control the spread of the disease worldwide by the use of booster mRNA vaccines; however, these have limited positive effects. The use of animal strains as live vaccines may be an alternative solution.

## Supporting information

**S1 Fig. Principal component 1 (PC1) of the African samples in Fig 3A observed in chronological order.** The upper third of the samples are cases with Omicron variant and the middle part is B.1.1.519. The earliest reports are from the uppermost part, which is the most mutated and infectious. The changes involved in the formation of the Omicron variant do not appear in this time series. Rather, it is more likely that a mature variant has entered the countries where these sequence testing methods are being carried out.
(TIF)

**S2 Fig. Simulation of a random mutation of a single variant: A random walk (dots).** The gray line is a sine curve, PC1 is half a cycle, and PC2 is one cycle. Influenza H1N1 showed a similar pattern. The length of the base was 1e4, and the number of trials was 1000. The average

of all results was used for centering before PCA. The results are scaled, but they are still much larger than those for COVID-19, which has not yet produced as many mutations.
(TIF)

**S3 Fig. Proportion of complete sequences registered in the GIS-AID per billion inhabitants in each region.** The axes are logarithmic. Africa is two orders of magnitude lower than the other regions.
(TIF)

**S4 Fig. Sequences of all Rodent coronaviruses from NCBI were compared with WHO-variant of concern (VOC).** Five samples of Omicron variant were used and the axes were set from all said samples. The rodent variants were very far from the human SARS-CoV-2; a difference that appears in principal component 1 (PC1). Only one SARS variant infecting rodents appeared somewhat closer to humans. The characteristics of the Omicron variant appear in PC 11, whereas the rodent virus does not show these characteristics at all. These viruses are unrelated.
(TIF)

**S5 Fig. WHO-variant of concern (VOC) axis with animal viruses.** Blue is the WHO-VOC. Each variant infected animals to a degree; the most recent.
(TIF)

## Acknowledgments

I would like to thank Editage (www.editage.com) for English language editing.

## Author Contributions

**Conceptualization:** Tomokazu Konishi.

**Data curation:** Tomokazu Konishi.

**Formal analysis:** Tomokazu Konishi.

**Funding acquisition:** Tomokazu Konishi.

**Investigation:** Tomokazu Konishi.

**Methodology:** Tomokazu Konishi.

**Project administration:** Tomokazu Konishi.

**Resources:** Tomokazu Konishi.

**Software:** Tomokazu Konishi.

**Supervision:** Tomokazu Konishi.

**Validation:** Tomokazu Konishi.

**Visualization:** Tomokazu Konishi.

**Writing – original draft:** Tomokazu Konishi.

**Writing – review & editing:** Tomokazu Konishi.

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
