## [Decision Letter · Decision Letter 0]

2 May 2022

PONE-D-22-05674Increasing SARS-CoV-2 mutations against vaccination-acquired immunityPLOS ONE

Dear Dr. Konishi,

Thank you for submitting your manuscript to PLOS ONE. After careful consideration, we feel that it has merit but does not fully meet PLOS ONE’s publication criteria as it currently stands. Therefore, we invite you to submit a revised version of the manuscript that addresses the points raised during the review process.

We look forward to receiving your revised manuscript.

Kind regards,

Paulo Lee Ho, Ph.D.

Academic Editor

PLOS ONE

Journal Requirements:

Reviewers' comments:

Reviewer's Responses to Questions

**Comments to the Author**

1. Is the manuscript technically sound, and do the data support the conclusions?

Reviewer #1: Yes

Reviewer #2: Yes

2. Has the statistical analysis been performed appropriately and rigorously? 

Reviewer #1: Yes

Reviewer #2: N/A

3. Have the authors made all data underlying the findings in their manuscript fully available?

Reviewer #1: Yes

Reviewer #2: Yes

4. Is the manuscript presented in an intelligible fashion and written in standard English?

Reviewer #1: Yes

Reviewer #2: Yes

5. Review Comments to the Author

Reviewer #1: Increasing SARS-CoV-2 mutations against vaccination-acquired immunity

- Tomokazu Konishi

Review comments:

Line number Comment

Line 16-17 “Monovalent vaccines using RNA or adenoviruses have successfully controlled the COVID-19 epidemic in many countries”

The above statement may be refined as “Monovalent vaccines using RNA or adenoviruses have provided substantial protection against the COVID-19 epidemic in many countries”

Line 18-19 The statement says the Omicron, “in particular”, put pressure on the healthcare system. But, weren’t hospitalizations comparatively lower for Omicron waves?

Line 26 Will “causing another pandemic wave” be more appropriate?

Line 79 Typo: Omicron variants

Line 162 Typo: …which are the surface proteins…

Is N considered a surface protein?

Line 215 Maybe the potential dangers can be specified

Line 224 But all viral proteins are not antigenic!

Line 217-228 How will the effectiveness of animal-adapted viral vaccines be different from currently available inactivated vaccines?

Is there any data to show the effectiveness/challenges/limitations of whole inactivated virus vs monovalent vaccines?

Line 246-253 Lines 229-236 is repeated in the Conclusion section

Reviewer #2: Manuscript #: PONE-D-22-05674

Title: Increasing SARS-CoV-2 mutations against vaccination-acquired immunity

Authors: Tomokazu Konishi

Article type: Research Article

The manuscript written by Dr. Tomokazu Konishi focused a major issue about COVID-19 booster vaccination specially to eradicate the effects of Omicron variants. The necessity of vaccination, about the commencement of herd immunity to prevent the mass public health from a pandemic are clearly stated in this manuscript. The objectives, results, and interpretations have been clearly given. Methods are sound as well. The knowledge gained from this work is of global interest. However, the manuscript needs language improvement. In some places, the superficial terms/ phrases have been used. If the manuscript is language edited and improved technically, it may warrant publication because of its merit and sound presentation quality. Therefore, the author is requested to fix some points as appended below.

In the Abstract section, the author may change the term epidemic to pandemic. All the VOCs should be mentioned. Also, the term “weaker variant” may be explained in a sentence in course of amino acid replacements/ deletions.

The Key Word section may add “booster dose”.

As a single author, in the Competing Interest section, “authors” cannot be used.

Introduction

Line 31: COVID-19 pandemic; this is not epidemic. Indeed, the first sentence in lines 31-32 is not clear and attractive. Please revise it.

Line 32: What does the author mean by the word “postulate”? Please use appropriate English.

Lines 37-38: Mention the potent variants with lineages with timeline in short. Give reference.

Line 39: How about the variants of interest (VOI)?

Line 46: That warning is a past tense indeed. Author must clearly discuss why the vaccines were in question in terms of their efficacy against the variants, especially against the Delta and Omicron variants. Then it was resolved by the application of the third dose; i.e., the so called booster dose. The memory B cells were activated by this booster. Author should put such explanation here with appropriate references.

Line 50: Omicron variant has two subtypes. Author should mention the names specifically with the number and position of mutations individually.

Line 53: Why “we”? Author himself is the author. Also, the manuscript should be written in passive form.

Lines 58-60: Author should take care of the “tense”. The language should be fixed with a native speaker as I have noticed so far after reading the Introduction section.

Materials and Methods

Line 63: Be more specific about the nucleotide sequence.

Line 77: What is “weighting errors”? Check the grammar.

Line 100: Fig. 2A shows NOT reflects

Lines 106-107: Give specific example of this sentence.

Line 114: What is meant by “each epidemic”?

Line 116: Author should also include the viral assortment in the H1N1 influenza.

Lines 121-122: I didn’t understand this sentence. Please write more specifically with immunological terms.

Lines 127-128: If the information is lacking, I think author should try to find out it. If he can’t, it’s better to delete this sentence. To my knowledge, no information is lacking regarding COVID-19 pandemic. Author should be more careful to include all the relevant data.

Line 149: Fix the typo error.

Lines 153-154: write as pandemic, and NOT epidemic.

Line 159: A simple sequence of mutations can be written here in case of the initial variant. Readers may understand well.

Line 163: Please address the mutations in Omicron variant quantitatively.

Lines 169-170: Re-write this sentence a simpler form. The word “farthest” is not a suitable one.

Line 175: Lambda and Delta ---- newer variants? Isn’t it contradictory? Please give explanation.

Discussion

Line 186: “discontinuous” in what sense? Author tried to explain that but this word is not suitable.

Line 200: Please include the frequency.

Line 202: “The USA and the UK are the most prolific sequencers.” --- This sentence is actually incomplete.

Line 209: How the author defines the monovalent vaccines? Author can try with the chemical composition of specific vaccines under trial.

Line 226: mRNA vaccines

Lines 231-233: Can author put some lines to explain the identical mutations in the Omicron variant?

Conclusion

Line 250-251: Weak sentence. I can’t get it actually.

Acknowledgement:

Line 257: Why “we”?

6. PLOS authors have the option to publish the peer review history of their article (what does this mean?). If published, this will include your full peer review and any attached files.

Reviewer #1: **Yes: **Suresh Thakur

Reviewer #2: **Yes: **Rashed Noor

---

## [Author Response · Author response to Decision Letter 0]

18 Jun 2022

Please see the atacched file, "ResponsetoReviewers.docx".

---

## [Decision Letter · Decision Letter 1]

28 Jun 2022

Mutations in SARS-CoV-2 are on the increase against the acquired immunity

PONE-D-22-05674R1

Dear Dr. Konishi,

We’re pleased to inform you that your manuscript has been judged scientifically suitable for publication and will be formally accepted for publication once it meets all outstanding technical requirements.

Kind regards,

Paulo Lee Ho, Ph.D.

Academic Editor

PLOS ONE

Additional Editor Comments (optional):

Reviewers' comments:

Reviewer's Responses to Questions

**Comments to the Author**

1. If the authors have adequately addressed your comments raised in a previous round of review and you feel that this manuscript is now acceptable for publication, you may indicate that here to bypass the “Comments to the Author” section, enter your conflict of interest statement in the “Confidential to Editor” section, and submit your "Accept" recommendation.

Reviewer #1: All comments have been addressed

Reviewer #2: All comments have been addressed

2. Is the manuscript technically sound, and do the data support the conclusions?

Reviewer #1: Yes

Reviewer #2: Yes

3. Has the statistical analysis been performed appropriately and rigorously? 

Reviewer #1: Yes

Reviewer #2: N/A

4. Have the authors made all data underlying the findings in their manuscript fully available?

Reviewer #1: Yes

Reviewer #2: Yes

5. Is the manuscript presented in an intelligible fashion and written in standard English?

Reviewer #1: Yes

Reviewer #2: Yes

6. Review Comments to the Author

Reviewer #1: Unlike the prevalent monovalent vaccine against highly mutated S protein, the usage of vaccine/s combining multiple epitopes/protein from other region may provide the better immunity against mutated variants.

Reviewer #2: Author has satisfactorily responded to my queries; and revised the manuscript accordingly. The revised form of manuscript is suitable for publication in Plos One.

7. PLOS authors have the option to publish the peer review history of their article (what does this mean?). If published, this will include your full peer review and any attached files.

Reviewer #1: **Yes: **Suresh Thakur Ph.D.

Reviewer #2: **Yes: **Rashed Noor

---

## [Editor Report · Acceptance letter]

1 Jul 2022

PONE-D-22-05674R1 

Mutations in SARS-CoV-2 are on the increase against the acquired immunity 

Dear Dr. Konishi:

I'm pleased to inform you that your manuscript has been deemed suitable for publication in PLOS ONE. Congratulations! Your manuscript is now with our production department. 

Kind regards, 

on behalf of

Dr. Paulo Lee Ho 

Academic Editor

PLOS ONE